# In Situ Regeneration of Copper-Coated Gas Diffusion Electrodes for Electroreduction of CO_2_ to Ethylene

**DOI:** 10.3390/ma14123171

**Published:** 2021-06-09

**Authors:** Magdalena Bisztyga-Szklarz, Krzysztof Mech, Mateusz Marzec, Robert Kalendarev, Konrad Szaciłowski

**Affiliations:** 1Academic Centre for Materials and Nanotechnology, AGH University of Science and Technology, al. A. Mickiewicza 30, 30-059 Krakow, Poland; mbs@agh.edu.pl (M.B.-S.); marzecm@agh.edu.pl (M.M.); 2Institute of Solid State Physics, University of Latvia, Kengaraga Street 8, LV-1063 Riga, Latvia; robert.kalendarev@cfi.lu.lv

**Keywords:** copper catalysts, GDE, carbon dioxide, ethylene, electrocatalysis, copper complexes

## Abstract

A key challenge for carbon dioxide reduction on Cu-based catalysts is its low faradic efficiency (FE) and selectivity towards higher-value products, e.g., ethylene. The main factor limiting the possibilities of long-term applications of Cu-based gas diffusion electrodes (GDE) is a relatively fast drop in the catalytic activity of copper layers. One of the solutions to the catalyst stability problem may be an in situ reconstruction of the catalyst during the process. It was observed that the addition of a small amount of copper lactate to the electrolyte results in increased Faradaic efficiency for ethylene formation. Moreover, the addition of copper lactate increases the lifetime of the catalytic layer ca. two times and stabilizes the Faradaic efficiency of the electroreduction of CO_2_ to ethylene at ca. 30%. It can be concluded that in situ deposition of copper through reduction of copper lactate complexes present in the electrolyte provides new, stable, and selective active sites, promoting the reduction of CO_2_ to ethylene.

## 1. Introduction

The increase of carbon dioxide concentration in the atmosphere is one of the major problems related to the greenhouse effect and will result in serious global warming issues [1]; it impacts the Earth’s surface energy balance and leads to severe environmental impacts [2]. A promising path to reduce carbon dioxide emissions into the atmosphere is to recycle carbon dioxide into fuels and commodity chemicals using electrochemical processes [3,4], especially using renewable energy sources, e.g., wind or solar [5]. Electrocatalytic conversion of CO_2_ into value-added chemicals and fuels is also important in the frame of the reduction of CO_2_ emissions [6,7]. Much research has been focused on copper as a metal catalyst because it is known to produce a mixture of methane, ethylene, and other gas products [8,9,10]. Due to chemical reactivity and the possibilities of further conversion (into epoxides and subsequently fine chemicals and polymers), ethylene is considered one of the most promising products of CO_2_ reduction [11]. Therefore, ethylene is sometimes called the king of petrochemicals because there are more commercial chemicals produced from ethylene than from any other intermediates, which is the result of several beneficial properties of ethylene as well as other technical and economic factors [12]. However, selectivity and Faradaic efficiency, as well as stability of the catalyst, need significant improvements. These may be achieved in a few ways, including minimalization of the overpotential for the carbon dioxide reduction reaction (CO2RR), an increase of catalyst selectivity, and an increase of the process efficiency.

Electrocatalytic reduction of renewable CO_2_ to hydrocarbons has a positive impact on the carbon balance in the atmosphere and may potentially help to reduce global warming issues, provided the application of a fully renewable energy source (vide supra). Advances in ethylene synthesis via catalytic CO_2_ electroreduction have made a significant step, from the first discovery of copper electrode catalytic properties, through advanced cell designs and selection of appropriate electrolytes. In practice, conversion of electrochemical carbon dioxide requires a catalyst capable of mediating the efficient formation of a single product with high selectivity at high current density. It turns out that maintaining high selectivity at high current densities remains a challenge. It has already been observed that a different copper electrode structure/morphology has a significant impact on the selectivity of CO_2_ reduction and faradaic efficiency of ethylene generation [13].

In addition, the process is promising for issues that overcome a number of the challenges facing the implementation of carbon-neutral energy sources because it provides a means of storage of renewable electric energy in a convenient, high-energy-density form [7]. Copper-based electrocatalysts have been extensively studied in various CO_2_ reduction processes. Depending on the form of the catalyst, electrolyte composition, and reaction conditions, various products, including carbon monoxide (CO) [14,15], mixtures of methane and ethylene [10,16,17], methanol (CH_3_OH) [18], and formic acid (HCOOH) [19] have been obtained. All listed reduction products may be formed through the electrochemical conversion of CO_2_ at the surface of Cu-based catalysts.

Electrocatalytic CO_2_ reduction is promising as a real process of CO_2_ utilization because it can occur under ambient conditions. The relatively low cost of copper (Cu) and its high catalytic activity towards the formation of hydrocarbons make it a very interesting catalytic material [9,20]. The application of copper in the processing of CO_2_ enables the formation of the C = C bond, making it especially interesting in the frame of the generation of ethylene, which is investigated in this work. Although long-chain hydrocarbons and a variety of products formed in minor quantities have been reported, methane and ethylene are predominant products of the electrochemical conversion performed with the use of Cu-based electrodes [9,21,22]. However, it should be underlined that the selectivity to any particular product is relatively low [23], and investigations focused on its increase are still of great importance. Furthermore, most of the studies of CO_2_ reduction to ethylene that show high Faradaic efficiency (FE) of over 70% are carried out at relatively low current density (<50 mA/cm^2^) or require porous polytetrafluroethene (PTFE) membrane [24] or Nafion coating [25] protecting the surface of the electrode. Such approach greatly increases electrode stability but at the same time may be considered as a significant drawback from the point of view of mass production and/or industrial applications.

Due to the diversity of obtained products as well as the importance of ethylene as a high-value feedstock chemical, it is desirable to increase the selectivity of copper for ethylene over methane. The ethylene selectivity also has broader consequences in the field of CO2RR catalysis because it provides insight into the C-C coupling reaction step that must occur to produce multi-carbon products. There have been many works published presenting the influence of copper structures on CO2RR. Here we report a simple way that can lead to a strategy targeting multi-carbon products—the addition of copper lactate to the electrolyte. Usually, electroreduction of carbon dioxide is performed at low current densities (<50 mA·cm^−2^) due to the limitation of mass transport [26]. A practical application would require, however, high-intensity processes; therefore, the current investigation is focused on improvements at a high current density regime (200 mA·cm^−2^) with the application of gas diffusion electrodes and a vigorous flow of electrolyte. It should be mentioned here that the application of gas diffusion electrodes (GDE) based on a highly porous carbon substrate with a thin catalytic copper layer results in geometric effects leading to changes in atomic arrangements at active sites. This in turn may result in product speciation significantly different from those observed for pure metal electrodes. Furthermore, gas diffusion electrodes reduce mass transfer limitations across the gas-liquid interface and to the catalyst surface.

The main problem limiting the possibilities of the long-term use of GDE-based catalysts is its relatively fast decrease in the amount of produced ethylene after a few minutes of electrolysis. Because the advantage of GDE’s is their ability to overcome mass transport limitations, the current density at which the system can operate without a significant loss of faradic efficiency (FE) is especially interesting. In this perspective, we present the possibility of modifying electrolysis conditions by adding a solution containing copper lactate complexes, which results in a longer lifetime of tested electrodes and increased activity and selectivity of CO2RR electrocatalysts, which may be of particular importance from the point of view of industrial applications. Increased Faradic efficiency of conversion of CO_2_ to ethylene is also important from a practical point of view because it is one of the most important building blocks for the chemical industry [27].

In this work, we focus on the electrocatalytic conversion of carbon dioxide to ethylene in an electrochemical flow reactor and the role of copper lactate complexes in this process. Here we present our recent investigations concerning catalytically active Cu@GDE electrodes and attempts to increase the lifetime of the electrocatalyst via in situ regeneration with copper lactate complexes. The stability of the electrocatalyst is evaluated in a direct way via product analysis, as well as with additional imaging with the use of scanning electron microscopy (SEM) and X-ray photoelectron spectroscopy (XPS). We postulate that the traces of soluble and stable copper complexes in the electrolyte restores the catalytic sites, promoting ethylene formation.

## 2. Materials and Methods

Our designed Cu-based catalytic coatings were deposited on functionalized surfaces, i.e., a gas diffusion layer (GDL) using PVD/PACVD (physical vapor deposition and plasma-assisted chemical vapor deposition) processes, which allows the production of well-defined coatings with the desired surface properties. A Cu thin film of 100 nm was deposited on a GDL sheet (Freudenberg H23C2: 140 × 100 mm^2^) by DC magnetron sputtering from a planar Cu target (purity 99.99%, dimensions 200 × 100 × 9 mm^3^) under argon atmosphere. The deposition process was performed using the vacuum coater G500M.1 (Sidrabe Vacuum, Ltd., Riga, Latvia). Before the deposition process, the chamber (≈ 0.1 m^3^) was pumped down to base pressure below 1.3 × 10^−5^ mbar by a turbo-molecular pump backed with a rotary pump. Then, a continuous Ar gas (purity 99.99%) flow of 20.0 cm^3^∙min^−1^ was introduced into the chamber. The pumping speed was altered by a throttle valve to set the sputtering pressure of 5.6 × 10^−3^ mbar. The target was sputtered in a constant DC mode at a power of 200 W and a voltage of 424 V. The distance between the target and the GDL sheet was approximately 17 cm, and the sheet was grounded and not heated intentionally during the deposition.

Electrochemical experiments were performed in MicroFlowCell (ElectroCell A/S, Tarm, Denmark), schematically shown in Figure 1. The experimental conditions for all GDE electrodes were: CO_2_ gas flow rate: 50 cm^3^∙min^−1^, electrochemically active surface area: 10 cm^2^. Original Ir-MMO anode (ElectroCell A/S, Tarm, Denmark) was used as a counter electrode. 1 M aqueous KHCO_3_ (Sigma–Aldrich, Saint Louis MO, USA, pure for analysis) was used as an electrolyte, electrolyte flow: c.a. 100 cm^3^∙min^−1^; the catholyte and anolyte were separated by a Nafion^®^ 117 membrane. Electrocatalytic tests were performed at galvanostatic conditions at a current density of 200 mA·cm^−2^ with the use of P211 potentiostat/galvanostat (Zahner Elektrik, GmbH, Kronach, Germany) working in two-electrode mode. The drop of copper lactate solution (1 mL) was injected into the electrolyte with a time interval of 30 min. The solution was prepared by the addition of 253.80 mL of 88% lactic acid (Chempur, Piekary Śląskie, Poland, pure for analysis) to 200 mL of deionized water (κ = 3.12 µS). Then, 99.88 g of CuSO_4_·5H_2_O (POCH, Gliwice, Poland, pure for analysis) was added to the solution and dissolved. In the next step, the pH of the solution was adjusted to 9.5 with the addition of NaOH (POCH, Gliwice, Poland, pure for analysis) and filled up to 1 L with deionized water at the continuous control of the pH value. A 3 M concentration of lactate in the solution at pH of 9.5 containing 0.4 M Cu^2+^ ensures complete complexation of all copper ions into the form of [Cu(HL)(L)]^–^ and [Cu(L)_2_]^2–^ complexes [28]. After injection of as-prepared copper lactate solution into a 1 M aqueous KHCO_3_ solution of pH = 8.16, the equilibrium is shifted towards [Cu(L)(HL)]^–^ and electrochemically inactive [Cu(HL)_2_] [28]; however, the concentration of the latter one is negligible. At these conditions, both speciation analysis [28,29] and UV-Vis spectroscopy indicate that the dominating form of copper is a monodeprotonated form, bis-lactatocopperate(II), [Cu(L)(HL)]—with a small contribution of neutral bis-lactato copper(II), [Cu(LH)_2_] (where LH represents the lactate monoanion). The volume of anolyte and catholyte were ~250 mL and ~1250 mL, respectively (cf. Figure 1a).

The measurements were conducted at room temperature and ambient pressure conditions. The qualitative and quantitative analysis of output gas was performed using GC-MS chromatograph QP2020 (Shimadzu, Kyoto, Japan). An MS line was equipped with a column (Agilent, J&W, 0.530 mm diameter, 30 m length), and helium with a gas flow rate of 377.5 mL·min^−1^ was used as carrier gas. A TCD line dedicated for analyses of hydrogen concentration was equipped with molecular sieve 5 Å packed column. In this case, nitrogen with a flow rate of 15 mL·min^−1^ was used as the carrier gas. The temperature of the oven during measurements was set at 30 °C. Before each measurement, both the molecular sieve and chromatographic column were conditioned at 150 °C for 600 s.

The Faradaic efficiency (FE) for several gas products was determined using Equation (1):(1)$$FE=\frac{V\cdot c\cdot z\cdot F\cdot p}{j\cdot A\cdot R\cdot T}\cdot 100\%$$
where *V*—gas flow at cell output in m^3^·s^−1^, *c*—concentration of the gaseous product in the output gas in%, *z*—the theoretical number of electrons consumed in formation of particular products (e.g., for C_2_H_4_ *z* = 12; for CH_4_ *z* = 8; for CO *z* = 2), *F*—Faraday constant (96485 C·mol^−1^), *p*—pressure (101 325 Pa), *j*—current density A·m^−2^, *A*—electrode geometric area in m^2^, *R*—gas constant (8.3144598 J·mol^−1^·K^−1^), *T*—temperature (298 K).

Quantitative analysis of electrode composition was performed using scanning electron microscopes (SEM) Quanta 3D 200i and Versa 3D (FEI) equipped with an EDAX energy dispersive X-ray spectroscopy (EDS) system with an energy resolution of 136 eV (for Mn Kα line).

Chemical states present at the surface of the electrodes before and after the electrocatalytic tests were investigated by X-ray photoelectron spectroscopy (XPS). The tests were performed with the use of the PHI Versa Probe II Scanning XPS system by applying a monochromatic Al Kα (1486.6 eV) X-ray focused to a 100 µm spot. The signal was collected from the surface of a 400 µm × 400 µm area.

## 3. Results

Electrocatalytic reduction of carbon dioxide to ethylene at copper-coated gas diffusion electrodes at high current density (cf. Experimental) results in the rapid decay of the catalytic layer. It is manifested in a color change of the electrode surface (Figure 2a,b), microstructural damages of the electrode surface (Figure 2c,d), dramatic changes of the electrode’s surface composition, as determined by the XPS spectroscopy (Figure 3), and decreased Faradaic yield of the process (cf. Figure 7). Naked-eye observations of the electrode surface have already revealed color change from reddish, copper-like colors to almost black, with only small yellowish-brown areas, indicating the disappearance or oxidation of the catalytic copper layer. Moreover, scanning electron microscopy observations indicate the presence of numerous micro-cracks formed at the electrode surface during the CO_2_ conversion process. Significant erosion of the electrode and increased porosity can be also easily noticed (Figure 2d).

XPS spectroscopy (Figure 3) indicates significant changes in the surface composition of the electrode, resulting from processes taking place at the electrode surface during electrocatalytic tests. An increase of signal intensity is seen at 286 eV, corresponding to the presence of the C-O bond at the electrode surface, which may indicate the presence of products of oxidation of the carbon substrate. The most visible change in recorded XPS spectra is the almost complete disappearance of the peaks corresponding to the presence of a copper signal at ~932 eV (Figure 3c). This indicates the almost complete loss of copper coating, which is consistent with naked-eye and microscopic observations of the surface of the tested electrode.

These preliminary data indicate oxidation (and even dissolution) of the catalytically active copper layer and even partial oxidative destruction of the carbon support (Appendix A). The dramatic decrease of the Cu2p peak in the XPS spectrum indicates almost complete removal of the copper catalyst from the electrode surface, whereas its shift towards higher energies, indicates copper oxidation. It has already been reported that copper nanostructures undergo rapid changes during the electroreduction of CO_2_; the degradation processes, along with poisoning, including Oswald ripening, reshaping, detachment, and dissolution [30], resulting in the formation of copper-carbonate complexes [31], is the most probable scenario. Observed oxide phases could have been formed after contact between the degraded electrodes and air, and it could also be due to decomposition of carbonate complexes during cell disassembly. Significant oxidative degradation at cathodic polarization may also be attributed, justified by high current density; therefore, there is local high concentration of highly reactive intermediate species of a free radical character, which may contribute to the oxidative degradation of electrodes [13]. Therefore, an attempt to regenerate in situ the electrocatalytic layer was undertaken. As the electrolysis is performed in a concentrated hydrogencarbonate solution of pH 7.86 (equilibration of 1 M KHCO_3_ electrolyte with CO_2_ results in a minute decrease of its pH), lactate copper complexes, known for their stability in a wide range of pH and their electrochemical reactivity, have been well described (Figure 4) [28,32].

At least three different families of copper complexes should be considered for these purpose: phosphonates, amino acid complexes, and carboxylates. The first family has been excluded on the basis of strong adsorption of phosphonates at copper surfaces, which could potentially lead to impaired electrocatalytic activity [33]. Furthermore, the common phosphonate baths used in copper electroplating yield smooth and uniform surfaces, which is highly undesired at the porous surfaces of gas diffusion electrodes. The high stability of amino acid complexes makes the electrodeposition of the catalytic layer rather questionable; furthermore, they tend to produce cuprous oxide deposits even at very negative potentials [34]. Among all carboxylate complexes, copper lactate and tartrate should be the most stable of the studied electrolytes. Out of these two, lactate seems to be the ligand of choice, as it should give copper depositing of the most porous structures (i.e., compatible with the surface of GDE) due to its tendency to induce electrochemical oscillations [35].

Under applied experimental conditions, both bis-lactatocopper(II) and its monodeprotonated form should be stable, whereas the presence of more bulky tetrakis-lactatocopperate(II) should be excluded (this complex is stable at a much higher pH) [29]. Both the [Cu(LH)_2_] and [Cu(L)(HL)]− complexes are not susceptible to nucleophilic attack by hydroxide, hydrogencarbonate, and carbonate ions at the metal center and are only engaged in a protolytic equilibrium involving the hydroxide group [32]. UV-vis analysis of the diluted stock solution of copper lactate confirms the stability of the complex (Figure 4). Copper lactate solution in 1 M KHCO_3_ shows an absorption maximum at 721 nm (711 nm in water at pH = 8.5), which is very close to the value of 700 nm reported for the [Cu(L)(HL)]− form [32]. This is consistent with speciation analysis, indicating the contribution of [Cu(LH)_2_] and [Cu(L)_2_]^2−^ below 10% at pH = 8. The hypsochromic shift upon dilution with pure water (final pH = 8.5) is consistent with the increase contribution of doubly deprotonated and weakly absorbing [Cu(L)_2_]^2−^ ions. A copper lactate electrolyte is commonly used to deposit copper(I) oxide, Cu_2_O. In the electrolyte saturated with CO_2,_ the equilibrium should be shifted towards the neutral [Cu(LH)_2_] complex; its primary electrolysis product is metallic copper. Moreover, at sufficiently low potentials, required for CO_2_ reduction, any Cu_2_O will be reduced to Cu [36]. Therefore, copper lactate can serve as an efficient copper source to regenerate the catalytically active surface of gas diffusion electrodes.

In our experiment, copper lactate complex addition was aimed at stabilizing the electrode, prevent its dissolution, and thereby improve its lifetime as well as increase the copper selectivity for ethylene over methane via regeneration of active sites at the electrode surface. No studies have yet been described in the literature in which the effect of the presence of copper-lactate complexes on the reduction of CO_2_ has been studied; however, a similar approach has been considered for the fabrication of copper-based cathodes for ethylene generation [37]. Top-view SEM images of Cu 100 nm coating deposited on Freudenberg H23C2 carbon paper are shown in Figure 5. It is observed that the thin Cu coating is present at the surface of the fibrous carbon substrate (Figure 5a). The electrodes before tests are fully and uniformly covered with copper with fine micro-cracks and micro-holes over the whole surface. After the electrocatalytic process (80 min) in pure 1 M KHCO_3_, only slight changes related to increased surface roughness were observed (Figure 5c,d). In contrast, after electrochemical tests in the electrolyte containing copper lactate (both for tests with one injection and a series of subsequent injections of copper lactate solution to the electrolyte), the morphology of the electrodes is different and is composed mainly from a large amount of various 3D structures—lichen-like scales (Figure 5e,f) and fine needle-like crystalline structures (Figure 5g,h) were identified at the surface. It is also seen that the degree of coverage with multidimensional structures increases with the amount of injected copper lactate complex.

EDS analysis of the electrodes (see Appendix A) indicated a very high concentration of carbon in comparison to copper, indicating that, during the electrochemical performance tests, copper was partially dissolved and the underlying carbon substrate become visible. Additionally, a high amount of oxygen suggests that Cu after tests existing mainly in the form of oxides and/or hydroxides. Moreover, localized point EDS analysis of specific 3D structures, visible in Figure 5e–h, indicates that areas are enriched with copper and oxygen, indicating the formation of copper oxides during electrocatalytic tests.

Changes in recorded XPS spectra for both electrolytes (1 M KHCO_3_ and 1 M KHCO_3_ + C_6_H_10_CuO_6_) indicate partial degradation of the electrode during the electrochemical tests (Figure 6). XPS analysis revealed the Cu2p peak at 934.0 eV and the satellite peak near 938−946 eV characteristic of Cu^2+^ species [38]. The absence of peaks associated with metallic Cu and Cu^+^ indicated that Cu^2+^ was the only copper form present at the surface of the GDE substrate. In addition, an increase in signal in the C1s range after the electrochemical test may be associated with an increase in the exposed area of the electrode due to copper dissolution. Furthermore, new signals, which have appeared at energies of 292 and 296 eV, were observed, indicating the presence of carbon-based compounds adsorbed at the electrode surface. The presence of oxygen may be associated with the rapid oxidation of copper, especially at active sites.

Injections of copper lactate solution to the catholyte during the electrocatalytic reduction of CO_2_ significantly modified both the composition of the product gas and the stability of the electrode. In Figure 7, we observe analyses of output gas composition for GDE without the addition of a complex (Figure 7a), with one injection of 1 mL of copper lactate stock solution (Figure 7b) and supplied with 1 mL of copper lactate stock solution every 30 min of the electrochemical tests (Figure 7c). The measurements were carried out at a constant current of 200 mA·cm^−2^. The injection of copper lactate solution increases the electrode lifetime.

In all studied cases, the main product of CO_2_ reduction is carbon monoxide. In the case of the pure KHCO_3_ electrolyte, ethylene is the second most abundant reduction product, but methane generation starts to dominate within the first 30 min of the process. The Faradaic yield of ethylene production reaches a maximum of ca. 40% in ca. 20 min. During the process, the Faradaic yield of ethylene gradually decreases, whereas the values for carbon monoxide and methane are approximately constant (Figure 7a). The same process, initiated with a single injection of copper lactate, is characterized by a significantly higher Faradaic efficiency of ethylene (almost 50% vs. 40% in the former case) and lower efficiencies of carbon monoxide and methane. Moreover, the decrease of ethylene yield is significantly slower (Figure 7b). In the whole process, ethylene is the CO_2_ reduction product with the highest Faradaic yield. A significant decrease of methane yield at the end of the process can be associated with the increase of cell voltage [24].

Finally, when the process is performed with sequential (1 mL/30 min) injections of copper lactate solution, the Faradaic yield of ethylene reaches 50% in the first 10 min but decreases and finally stabilizes at the level of 30% (Figure 7c). The amount of carbon monoxide is slightly higher than in the former case. It should be noted that hydrogen is always the most abundant product of electrolysis due to high cell voltage.

The shift of product distribution from CO to hydrocarbons (Figure 7a–c) is attributed to the activity of newly formed active sites, but the presence of lactate anions, which may adsorb at the surface of the cathode may also slightly modulate the product distribution. Furthermore, GC analysis showed significantly higher electrode selectivity for ethylene production than methane after modifying the composition of a basic electrolyte by copper lactate complexes.

Sequential supplementation with small aliquots of copper lactate stock solution results in a significant increase of electrocatalyst lifetime and increased FE of ethylene production (Figure 7c). This process seems to regenerate (or generate new sites of different structure but similar activity) catalytically active sites. This leads to a twofold increase of the electrode lifetime as well as improved selectivity towards ethylene. This is confirmed by SEM and XPS measurements. Addition of copper lactate to the catholyte results in the deposition of copper-based nanostructures that seem to be at least as active as freshly sputtered copper surface. Furthermore, at the time of the addition of C_6_H_10_CuO_6_, we observed a slight and temporary decrease in the amount of CO (Figure 7c), which confirms this thesis. In the electrolyte supplied with 1 mL of copper lactate solution at a 30-min interval, the initial stage of the experiment was similar to that which was carried out with a single addition of copper lactate complex. Its further addition did not significantly increase the amount of C_2_H_4_ in the output gas but provided conditions in which ethylene FE decreased over time slower than in the case without copper complex addition. In addition, it has been observed that the addition of copper lactate increases the selectivity towards ethylene production. Unfortunately, after about 90 min (Figure 7), we observed a significant increase in cell potential. This can be associated with the migration of ions from the electrolyte through the membrane. Nafion should act as a proton conductor, but at the high current densities used in the reported experiments, it also undergoes gradual degradation, which is manifested by significant amounts of fluoride anion detected at the electrode surfaces (Appendix A).

In the electrolyte supplied with 1 mL of copper lactate solution applying 30 min time interval, the initial stage of the experiment proceeded similarly to those performed in the solution containing copper lactate complexes at the beginning (Figure 7c). Further additions of copper lactate did not enhance significantly the quantity of the C_2_H_4_ produced in the output gas, but it did provide the limiting conditions at which FE for ethylene does not decrease with time. Moreover, it was observed that the addition of copper lactate enhances the selectivity towards ethylene generation. This fact could be a significant step in further optimization of GDE-based conversion of carbon dioxide. Unfortunately, high current density results in low cathode potential, and along with the CO_2_ reduction, hydrogen production occurs with significant Faradaic efficiency.

To summarize, the addition of copper lactate to the basic electrolyte has proven to be a simple, fast, and direct method, leading to both further increasing the electrode lifetime and enhancing the C2 product selectivity.

## 4. Conclusions

In summary, we presented a new strategy for directly extending the lifetime of GDE-based Cu catalysts to investigate CO2RR performance at high current densities by the simple, fast, and cheap method of the addition of copper lactate solution to 1 M KHCO_3_ electrolyte. The addition of copper lactate complex, presented in this paper, leads to increased selectivity for ethylene over methane. Due to the increase of selectivity towards ethylene production after the addition of these complexes, it is reasonable to conclude that the addition of copper lactate contributes to the formation of new catalytically active sites that take part in the selective reduction of carbon dioxide to ethylene. The nature of the new catalytic sites is not recognized yet and will be the subject of further study. Possibly the large active surface area of fresh copper deposits, as well as the possibility of the adsorption of lactate anions, contribute to the observed enhancement of the catalytic performance of GDE cathodes in the electrochemical reduction of carbon dioxide to ethylene.

Although these results are very promising, they can only serve as a basis for further research. Importantly, further work must examine the feasibility of moving this approach from semi-periodic to continuous mode and assess the long-term stability of GDE performance.

## Figures and Tables

**Figure 1 materials-14-03171-f001:**
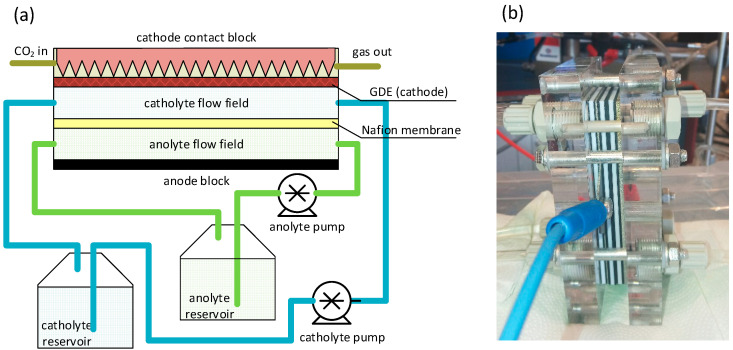
Scheme of the electrochemical GDE-based flow cell setup, including a cross-section of the cell (**a**) and a real photo of the cell (**b**).

**Figure 2 materials-14-03171-f002:**
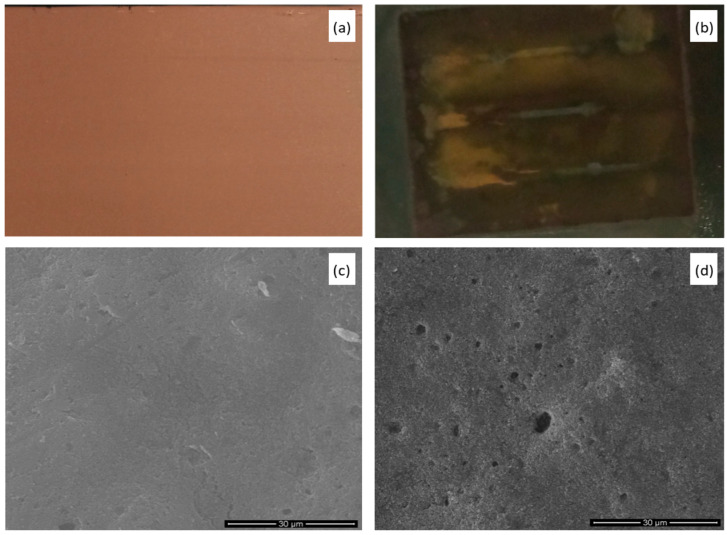
Photographic (**a**,**b**) and SEM (**c**,**d**) images of copper-sputtered GDE before (**a**,**c**) and after (**b**,**d**) 80 min-long electrocatalytic tests performed at the current density of 200 mA·cm^−2^.

**Figure 3 materials-14-03171-f003:**
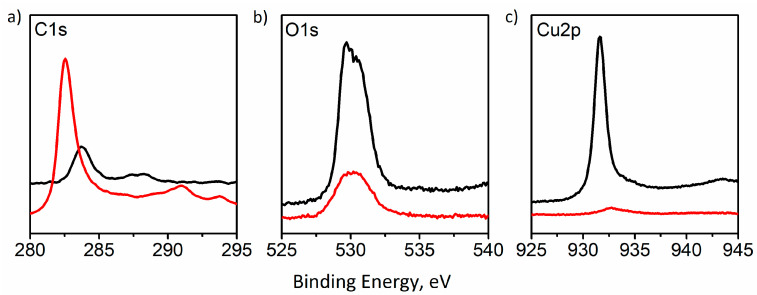
XPS spectra recorded for the GDE surface before (black curve) and after (red curve) 80 min of electrolysis at the current density of 200 mA·cm^−2^. (**a**):C1s; (**b**):O1s; (**c**):Cu2p.

**Figure 4 materials-14-03171-f004:**
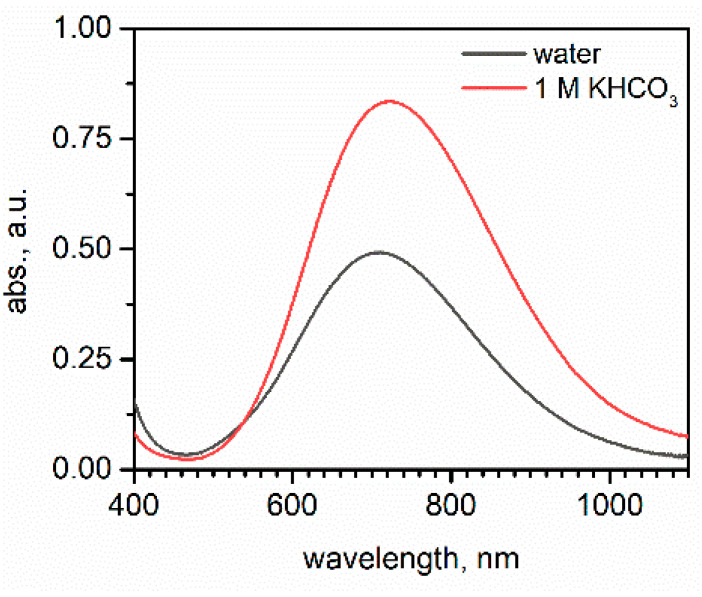
Electronic spectra of copper lactate complexes (16 mM) in water (pH = 8.5) and 1 M KHCO_3_ solution (pH = 8.16).

**Figure 5 materials-14-03171-f005:**
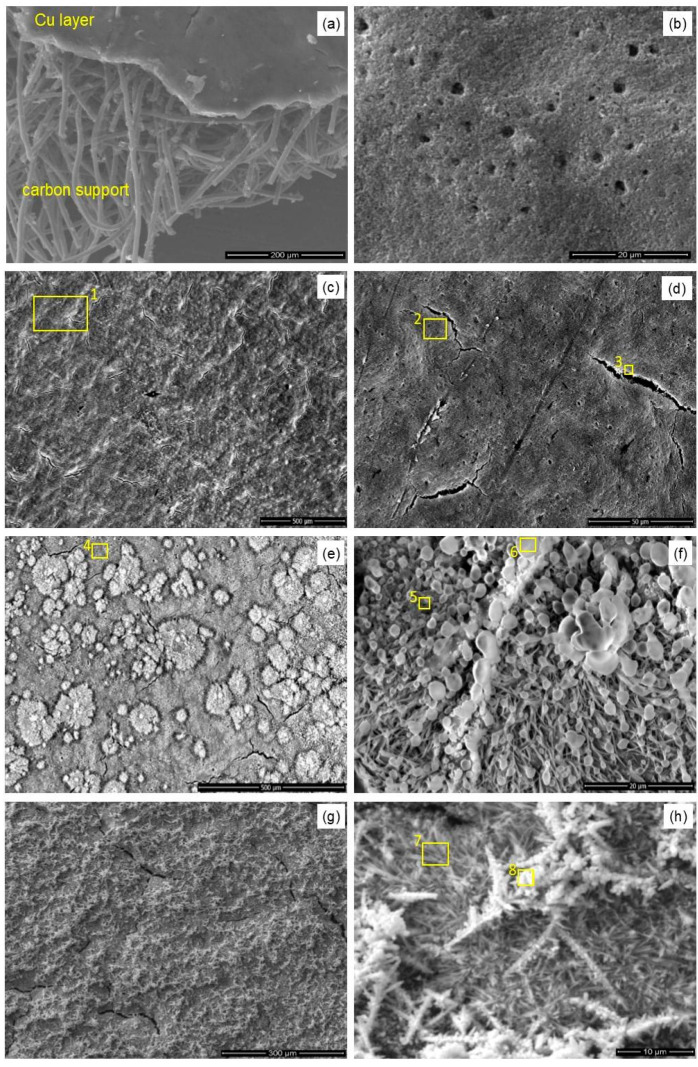
SEM images of Cu 100 nm sputtered on Freudenberg H23C2 carbon paper before the process (**a**,**b**), after 1-h test in 1M KHCO_3_ (**c**,**d**), after 1-h test in 1M KHCO_3_ with a single addition of 1 mL of copper lactate solution, (**e**,**f**), and after a test with a series of copper lactate injections (**g**,**h**) (200 mA·cm^−2^). See Appendix A in ESI for the details of EDS analyses of labeled areas.

**Figure 6 materials-14-03171-f006:**
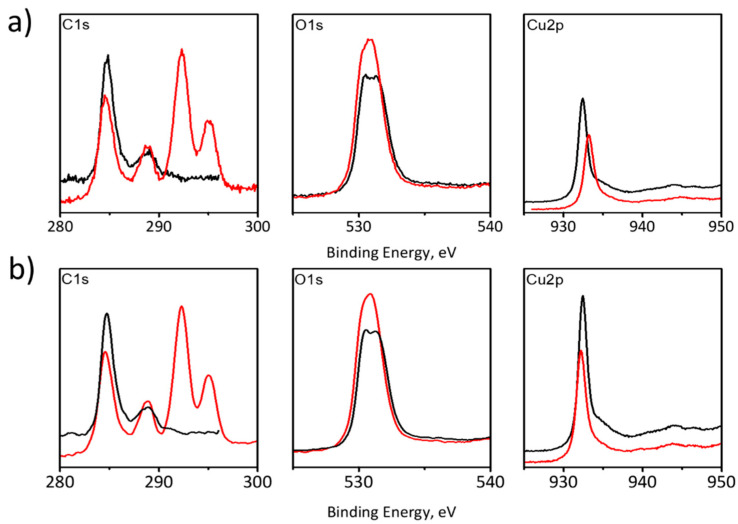
XPS analysis of Cu-electrode before (black curve) and after (red curve) tests in (**a**) 1 M KHCO_3_ and (**b**) 1 M KHCO_3_ + C_6_H_10_CuO_6_.

**Figure 7 materials-14-03171-f007:**
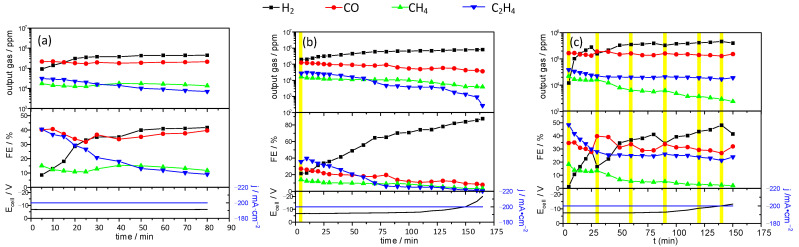
Output gas composition and Faradaic efficiency of CO_2_ conversion for: (**a**) without C_6_H_10_CuO_6_ addition; (**b**) single C_6_H_10_CuO_6_ addition at the beginning of the experiment, and (**c**) the effect of the multiple C_6_H_10_CuO_6_ additions with ca. 30 min time intervals (200 mA·cm^−2^, 1 M KHCO_3_). Each copper lactate injection is marked as a yellow bar.

## Data Availability

Data concerning the several measurements are available upon request.

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
