# Peer review of "In Situ Regeneration of Copper-Coated Gas Diffusion Electrodes for Electroreduction of CO2 to Ethylene"

_materials, 2021, doi:10.3390/ma14123171_

Round 1

Reviewer 1 Report

In this review manuscript authors present role of copper lactate in electrocatalytical conversion of CO2, in order to improve selectivity of method toward specific product (ethylene) and increasing of lifetime of the catalyst.

The manuscript is well organized and supported by experimental data, however, some points in the manuscript left meagre explained. So, the authors need to provide some additional explanation or experimental data that occurred during measurements:

  1. First, one general question: Electrochemical reduction of CO2 take place on a porous solid electrode. In which form is CO2 at the vicinity of the electrode? In the form of gas or as dissolved in electrolyte?

  1. Inserting part of the manuscript is the fact that after cathodic reduction of CO2, copper is in the form of oxides or hydroxide. Authors assign this behaviour to the oxidation of copper. However, more insight in the process that takes place at porous copper electrode would be provided by data of the change of the potential at the porous copper electrode during chronoamperometric measurements.

  1. Also, many other complexing agens can be used in this principle, such as phosphonate (pH 8-8.5) or phosphonic acid. The manuscript would provide more relevant and comparable information about efficiency if other complex agens were a point of interest.

SPECIFIC COMMENTS:

lines 267-268: It seems that instead of "Figures 6e-h" there should be 5e-h.

Author Response

On behalf of all authors I would like to thank the reviewer for constructive comments. All comments have been taken info account and the manuscript has been revosed accordingly.

  1. First, one general question: Electrochemical reduction of CO2 take place on a porous solid electrode. In which form is CO2 at the vicinity of the electrode? In the form of gas or as dissolved in electrolyte?

In this study have been used carbon paper based gas diffusion electrodes. In the vicinity of the electrode CO2 is in the gaseous form, the reaction takes place at the ternary gas-liquid-solid interface.

  1. Inserting part of the manuscript is the fact that after cathodic reduction of CO2, copper is in the form of oxides or hydroxide. Authors assign this behaviour to the oxidation of copper. However, more insight in the process that takes place at porous copper electrode would be provided by data of the change of the potential at the porous copper electrode during chronoamperometric measurements.

The processes responsible for degradation has been described in more detail and supported with recent literature.

  1. Also, many other complexing agents can be used in this principle, such as phosphonate (pH 8-8.5) or phosphonic acid. The manuscript would provide more relevant and comparable information about efficiency if other complex agents were a point of interest.

A detailed justification of the choice of ligands has been presented in the revised manuscript.

SPECIFIC COMMENTS:

lines 267-268: It seems that instead of "Figures 6e-h" there should be 5e-h.

We are very sorry for confusion, the mistype has been corrected.

Reviewer 2 Report

This is an interesting research considering the efficiency of copper regeneration at gas diffusion electrode applied for electrochemical conversion of CO2 to ethylene. The authors have shown that periodically addition of copper lactate complexes improved the yield of target product. Unfortunately, the description of the key results presented in Figure 7 contradict with the graphs presented and the captions given. This complicates the assessment of the results. I would suggest first to correct the obvious misprints and mistakes through the text and the return to the manuscript reviewing.  IN addition to the Fig. 7 problems, the authors should remove the duplications in the text describing the importance of ethylene production from carbon dioxide and carefully check the use of acronyms, especially FE which was introduced in the middle of manuscript but not used prior to and after such implementation.

Technical notes:

Abstract: First and second sentences partially duplicate each other.

Introduction, lines 44,45: Here, “reduction” is used within one sentence in two different senses – as “decrease” and as one of redox reactions – please rephrase to avoid confusing

line 93: GDE acronym should be introduced earlier, at line 89, where the term ‘gas diffusion electrode’ appeared for the first time

Line 99 – FE acronym has not been jet introduced

Line 112: please introduce the acronyms SEM and XPS

Line 167: Please check the Eq.(1) – there is no Faraday constant in the formula and the pressure is probably expressed in atmospheres, not Pa

Line 188: Please check the link to Figure 4 – it does not contain information on Faradaic yield

Line 215 “hydrogen carbonate” should be replaced with “hydrocarbonate”

Line 218: Please check if Figure 4 or 5 is mentioned

Figure 7 – Please check the legend, there is no ethylene (C2H2 is acetylene) The description of the Figure 7 done in the lines 297-307 contradicts with appropriate figures. Please check if the captions are correct.

Author Response

On behalf of all authors I would like to thank the reviewer for constructive comments. All comments have been taken info account and the manuscript has been revosed accordingly.

This is an interesting research considering the efficiency of copper regeneration at gas diffusion electrode applied for electrochemical conversion of CO2 to ethylene. The authors have shown that periodically addition of copper lactate complexes improved the yield of target product.

Unfortunately, the description of the key results presented in Figure 7 contradict with the graphs presented and the captions given.

Figure caption has been corrected as requested

This complicates the assessment of the results. I would suggest first to correct the obvious misprints and mistakes through the text and the return to the manuscript reviewing. In addition to the Fig. 7 problems, the authors should remove the duplications in the text describing the importance of ethylene production from carbon dioxide and carefully check the use of acronyms, especially FE which was introduced in the middle of manuscript but not used prior to and after such implementation.

The manuscript has been revised as requested. Equation 1 was checked as requested,

Technical notes:

Abstract: First and second sentences partially duplicate each other.

Corrected as requested

Introduction, lines 44,45: Here, “reduction” is used within one sentence in two different senses – as “decrease” and as one of redox reactions – please rephrase to avoid confusing

Corrected as requested

line 93: GDE acronym should be introduced earlier, at line 89, where the term ‘gas diffusion electrode’ appeared for the first time

Corrected as requested

Line 99 – FE acronym has not been jet introduced

Corrected as requested

Line 112: please introduce the acronyms SEM and XPS

Corrected as requested

Line 167: Please check the Eq.(1) – there is no Faraday constant in the formula and the pressure is probably expressed in atmospheres, not Pa

The Faraday constant is assigned by F. The pressure should be expressed in Pa.

Line 188: Please check the link to Figure 4 – it does not contain information on Faradaic yield

Corrected as requested

Line 215 “hydrogen carbonate” should be replaced with “hydrocarbonate”

We are sorry, mistype is corrected, the preferred IUPAC name HYDROGENCARBONATE is used in the text.

Line 218: Please check if Figure 4 or 5 is mentioned

Corrected as requested

Figure 7 – Please check the legend, there is no ethylene (C2H2 is acetylene) The description of the Figure 7 done in the lines 297-307 contradicts with appropriate figures. Please check if the captions are correct.

Corrected as requested

Reviewer 3 Report

This manuscript shows one way to regenerate the active sites for electrochemical CO2 reduction by adding some copper lactate to the electrolyte. This idea seems very interesting, and this manuscript can be a good supplement to the electrochemical CO2 reduction research community. Please find my questions and concerns below.

  1. In the Introduction part, the authors claimed that “It has already been observed that a different copper electrode structure/morphology has a significant impact on the selectivity of CO2 reduction and faradaic efficiency of ethylene generation.” I think the audience including me would expect some brief explanations and some literature cited here.

  1. Figure 1 is apparently very confusing. The cell holds much more catholyte than anolyte (based on the method description), but the figure shows that the volume of catholyte and anolyte is the same. Does the electrolyte have direct contact with the atmosphere? This is what I got from the figure. I understand it’s only a schematic figure, but it at least should be representative and precise.

  1. This is minor. There are some typos and grammar issues that need to be taken care of carefully. For instance, “In all studied cases the main product of CO2 reduction of carbon monoxide” should be “In all studied cases the main product of CO2 reduction is carbon monoxide” in line 297.

  1. In Figure 7a, methane production looks very stable, but in Figure 7b methane productions dropped dramatically (to almost 0) after 100 min. How do you explain this?

  1. After adding copper lactate, did you do the long-term testing (e.g. 24 hours)? I can only see 100 mins of testing from Figure 7b. If the newly generated copper sites are still not stable and can last only 20 mins, I don’t think this is a practical way to “regenerate” the electrode. Besides, what is the structure of your newly generated copper sites? It looks to me that the new sites are not the same with the original ones. If this is true, I don’t think you can claim it is a “regeneration” of the active sites.

  1. What is the state-of-art production rate or FE of ethylene for electrochemical CO2 reduction? Could you show some comparisons of your studies with other studies? Based on my knowledge, your ethylene production seems to be lower than other studies.

  1. I didn’t find any electrochemical characterizations of your electrode and electrolyte, such as the Cyclic Voltammetry curve, which is a very typical test for this kind of topic.

Author Response

On behalf of all authors I would like to thank the reviewer for constructive comments. All comments have been taken info account and the manuscript has been revosed accordingly.

  1. In the Introduction part, the authors claimed that “It has already been observed that a different copper electrode structure/morphology has a significant impact on the selectivity of CO2 reduction and faradaic efficiency of ethylene generation.” I think the audience including me would expect some brief explanations and some literature cited here.

A reference to a recent review has been added as requested.

  1. Figure 1 is apparently very confusing. The cell holds much more catholyte than anolyte (based on the method description), but the figure shows that the volume of catholyte and anolyte is the same. Does the electrolyte have direct contact with the atmosphere? This is what I got from the figure. I understand it’s only a schematic figure, but it at least should be representative and precise.

New figure 1, showing the whole setup, has been used.

  1. This is minor. There are some typos and grammar issues that need to be taken care of carefully. For instance, “In all studied cases the main product of CO2 reduction of carbon monoxide” should be “In all studied cases the main product of CO2 reduction is carbon monoxide” in line 297.

Corrected

  1. In Figure 7a, methane production looks very stable, but in Figure 7b methane productions dropped dramatically (to almost 0) after 100 min. How do you explain this?

Methane production was of minor importance from the point of view of the goals of our research, so it was not analyzed in detail. We can, however hypothesize, that this is associated with a significant increase of cell voltage near the end of the experiment. Appropriate comments has been added to the text.

  1. After adding copper lactate, did you do the long-term testing (e.g. 24 hours)? I can only see 100 mins of testing from Figure 7b. If the newly generated copper sites are still not stable and can last only 20 mins, I don’t think this is a practical way to “regenerate” the electrode. Besides, what is the structure of your newly generated copper sites? It looks to me that the new sites are not the same with the original ones. If this is true, I don’t think you can claim it is a “regeneration” of the active sites.

We agree with the reviewer. The structure of newly generated copper catalytic sites may be different than the original ones, generated by sputtering. Appropriate comment has been added to the manuscript.

  1. What is the state-of-art production rate or FE of ethylene for electrochemical CO2 reduction? Could you show some comparisons of your studies with other studies? Based on my knowledge, your ethylene production seems to be lower than other studies.

The state of the art is briefly discussed on the basis of newest literature repots.

  1. I didn’t find any electrochemical characterizations of your electrode and electrolyte, such as the Cyclic Voltammetry curve, which is a very typical test for this kind of topic.

CV curves would be useful for analysis of stationary solutions, current study has been performed under vigorous flow, therefore these measurements have been abandoned.

Round 2

Reviewer 2 Report

I am satisifed with the changes made and suggest to accept the manuscirpt in present form

Reviewer 3 Report

I don't think my concerns are fully addressed but this manuscript can be accepted.